# FBXW5 Promotes Tumorigenesis and Metastasis in Gastric Cancer via Activation of the FAK-Src Signaling Pathway

**DOI:** 10.3390/cancers11060836

**Published:** 2019-06-17

**Authors:** Mei Shi Yeo, Vinod Vijay Subhash, Kazuto Suda, Hayri Emrah Balcıoğlu, Siqin Zhou, Win Lwin Thuya, Xin Yi Loh, Sriganesh Jammula, Praveen C. Peethala, Shi Hui Tan, Chen Xie, Foong Ying Wong, Benoit Ladoux, Yoshiaki Ito, Henry Yang, Boon Cher Goh, Lingzhi Wang, Wei Peng Yong

**Affiliations:** 1Department of Haematology-Oncology, National University Hospital of Singapore, Singapore 119228, Singapore; e0222965@u.nus.edu (M.S.Y.); shihuitan.14@hotmail.com (S.H.T.); matchxie@hotmail.com (C.X.); wf_ying@hotmail.com (F.Y.W.); boon_cher_goh@nuhs.edu.sg (B.C.G.); 2Cancer Science Institute of Singapore, National University of Singapore, Singapore 117599, Singapore; csiks@nus.edu.sg (K.S.); siqin_5@hotmail.com (S.Z.); csithuya@nus.edu.sg (W.L.T.); csiloh@nus.edu.sg (X.Y.L.); csipcp@nus.edu.sg (P.C.P.); csiitoy@nus.edu.sg (Y.I.); csiyangh@nus.edu.sg (H.Y.); 3Lowy Cancer Research Centre, University of New South Wales, Sydney 20152, Australia; 4Mechanobiology Institute, National University of Singapore, Singapore 117411, Singapore; balciemrah@gmail.com (H.E.B.); benoit.ladoux@ijm.fr (B.L.); 5Cancer Research UK Cambridge Institute, University of Cambridge, Cambridge CB2 0RE, UK; Sriganesh.Jammula@cruk.cam.ac.uk; 6Institut Jacques Monod, Centre National de la Recherche Scientifique, CNRS UMR 7592, Université Paris-Diderot, CEDEX 13, 75205 Paris, France

**Keywords:** FBXW5 (F-box/WD repeat-containing protein 5), tumorigenesis, metastasis, FAK-Src signaling

## Abstract

F-box/WD repeat-containing protein 5 (FBXW5) is a member of the FBXW subclass of F-box proteins. Despite its known function as a component of the Skp1-Cullin-F-box (SCF) ubiquitin ligase complex, the role of FBXW5 in gastric cancer tumorigenesis and metastasis has not been investigated. The present study investigates the role of FBXW5 in tumorigenesis and metastasis, as well as the regulation of key signaling pathways in gastric cancer; using in-vitro FBXW5 knockdown/overexpression cell line and in-vivo models. In-vitro knockdown of FBXW5 results in a decrease in cell proliferation and cell cycle progression, with a concomitant increase in cell apoptosis and caspase-3 activity. Furthermore, knockdown of FBXW5 also leads to a down regulation in cell migration and adhesion, characterized by a reduction in actin polymerization, focal adhesion turnover and traction forces. This study also delineates the mechanistic role of FBXW5 in oncogenic signaling as its inhibition down regulates RhoA-ROCK 1 (Rho-associated protein kinase 1) and focal adhesion kinase (FAK) signaling cascades. Overexpression of FBXW5 promotes in-vivo tumor growth, whereas its inhibition down regulates in-vivo tumor metastasis. When considered together, our study identifies the novel oncogenic role of FBXW5 in gastric cancer and draws further interest regarding its clinical utility as a potential therapeutic target.

## 1. Introduction

Gastric cancer is the fifth most commonly diagnosed malignancy and the third leading cause of cancer-related death worldwide [1]. Although the worldwide incidence of gastric cancer has declined remarkably over the recent few decades [2,3,4], disease burden remains high and gastric cancer is traditionally associated with poor prognosis. Poor gastric cancer prognosis is attributed to the late diagnosis in most patients. This is because most patients are asymptomatic in the early stage [5], with most cases often diagnosed in the late stages when metastatic gastric cancer has developed. Statistics have shown that 79% of patients are diagnosed at stage IV, with a less than 5% five-year survival rate. Like other cancer metastases, gastric cancer metastasis is a serious condition whereby the median overall survival does not exceed one year [6].

An increase in focal adhesion kinase (FAK) activity or overexpression of FAK has been reported to give rise to an aggressive tumor phenotype in gastric cancer. The activation of FAK has been widely reported to promote in-vitro cell proliferation and migration, as well as in-vivo tumorigenesis and metastasis in gastric cancer [7,8,9,10,11]. The occurrence of these phenotypic changes has been determined to be driven by the activation of downstream signaling pathways, such as Src/ERK [9,12] or STAT3/NF_K_β [10] in response to an increase in p-FAK expression. Studies have also reported that RhoA-ROCK1 (Rho-associated protein kinase 1) signaling regulates the formation of focal adhesions to promote the activation of FAK, which in turn plays a crucial role in actin cytoskeleton remodeling and focal adhesion turnover to promote the metastasis of cancer cells [13].

A role of F-box protein in FAK-dependent proliferation of endothelial cells has been shown previously [14]. Despite its known function as a component of substrate degradation complexes, the role of F-box proteins in oncogenic signaling and cancer metastasis is undefined. The human F-box/WD repeat-containing protein 5 (FBXW5) is a member of the FBXW (F-box and WD40) subclass of F-box proteins [15], which acts as the substrate recognition component of both SCF (SKP1-CUL1-F-box protein) and DCX (DDB1-CUL4-X-box) E3 ubiquitin-protein ligase complexes [16]. There is emerging evidence to suggest that dysregulated degradation of tumor suppressors and oncoproteins by F-box proteins can drive tumorigenesis [17,18,19]. A recent study reported the growth promoting effect of FBXW5 in non-small cell lung cancer (NSCLC), via the degradation of Deleted in Liver Cancer (DLC1), a Rho GTPase-activating protein tumor suppressor [20]. Consistent with this, a functional role of FBXW5 in cell proliferation was also reported in HeLa cells [21]. Collectively, these studies point towards a potential oncogenic role of FBXW5 that warrants further investigations. In the current study, we investigated the role of FBXW5 in the regulation of tumorigenesis and metastasis, as well as its effect on the FAK-Src pathway. Our study demonstrates a novel oncogenic role of FBXW5 in gastric cancer and draws further interest regarding its clinical utility as a potential prognostic biomarker and therapeutic target.

## 2. Results

### 2.1. FBXW5 Knockdown Decreases Cell Proliferation and Increases Apoptosis in Gastric Cancer

Four gastric cancer cell lines, MKN1, CLS145, AGS and SNU1, were shown to have varying levels of FBXW5 mRNA and protein expressions (Appendix AA,B). Subsequently, MKN1 was used as an in-vitro model, considering its high mRNA expression of FBXW5. FBXW5 expression was knocked down in MKN1 cells using FBXW5 targeted siRNAs to investigate the role of FBXW5 in gastric cancer cell proliferation (Figure 1A,B).

Knockdown of FBXW5 resulted in a significant reduction in cell proliferation (*p*-values = 0.01, 0.0008 and 1.74 × 10^−6^, respectively) (Figure 1C), with a significant increase in sub G1 and G0/G1 phase cells (*p*-values = 0.02 and 0.04, respectively) and a corresponding decrease in S phase cells (*p*-value = 0.01) (Figure 1D). Knockdown of FBXW5 also led to an increase in cellular apoptosis. Here, a two-fold increase in Annexin V positive cells (*p*-value = 0.03) (Figure 1E) and a significant increase in caspase-3 activity (*p*-value = 0.03) (Figure 1F) were observed in MKN1 FBXW5 KD cells as compared to its non-targeting control. The role of FBXW5 in apoptotic regulation was further corroborated by Western blot analyses that showed significant down regulation of survivin and up regulation of p21 expressions in MKN1 FBXW5 KD cells as compared to MKN1 NT cells (Figure 1G).

### 2.2. FBXW5 Knockdown Down Regulates Actin Dynamics via Rho Signaling Pathway

The effect of FBXW5 knockdown on cell migration was determined. As compared to the non-targeting control, which remained highly migratory, FBXW5 depleted MKN1 cells exhibited reduced migratory potential (Figure 2A). We also investigated the effect of FBXW5 overexpression on cell migration using a stable expression of Myc-DDK-tagged-Human FBXW5 cDNA cloned in MKN1 cells (Figure 2B,C). Here, MKN1 FBXW5 OE (overexpression) cells were observed to have greater migratory potential than MKN1 control cells (Appendix AA). Since FBXW5 was observed to have a role in regulating cell migration, an immunofluorescence staining of F-actin was performed to determine changes within the actin cytoskeleton. Loss of FBXW5 caused cytoskeletal rearrangement and depolymerization of actin stress fibers, whereas the overexpression of FBXW5 increased polymerization of actin stress fibers, as shown by a decrease in F-actin staining in the MKN1 FBXW5 KD cells and an increase in F-actin staining in the MKN1 FBXW5 OE cells, respectively (Figure 2D). The role of FBXW5 in actin polymerization was further validated by Western blot analyses that showed a significant reduction in F-actin to total (G+F) actin protein expression, with no significant change in G-actin, in MKN1 FBXW5 KD cells as compared to MKN1 NT cells (Figure 2E). Subsequently, we investigated the effect of FBXW5 knockdown on RhoA-ROCK1-pMLC2 signaling and the activities of Cdc42 and Rac1. Western blot analyses showed a significant reduction in RhoA, ROCK1 and pMLC2 expressions in MKN1 FBXW5 KD cells as compared to MKN1 NT cells (Figure 2F). Knockdown of FBXW5 also decreased Cdc42 activity as represented by the down regulation in Cdc42-GTP protein expression (Figure 2G), whereas no change in Rac1 activity was observed (Appendix AB).

### 2.3. FBXW5 Knockdown Decreases Focal Adhesion Turnover and Down Regulates Cell Adhesion

Since the role of focal adhesions in actin polymerization and cytoskeletal rearrangement was established previously [22], we performed an immunofluorescence staining of phospho-paxillin to visualize the distribution and turnover of focal adhesions. Knockdown of FBXW5 drastically decreased the distribution of focal adhesions, as demonstrated by the sparse phospho-paxillin expression at the cellular boundaries (Figure 2H, left panel). Conversely, a rescued phenotype was observed in MKN1 FBXW5 KD cells with the stabilization of focal adhesions upon treatment with nocodazole (Figure 2H, middle panel), a microtubule-depolymerizing agent that reversibly blocks focal adhesion disassembly [23,24,25]. The decrease in focal adhesion turnover as a result of FBXW5 knockdown was demonstrated by a complete reversal of the previous observation, whereby a significant decrease in focal adhesion distribution was once again observed following the wash out of nocodazole (Figure 2H, right panel). On the contrary, no significant changes in phospho-paxillin staining were observed in the MKN1 NT cells before and after exposure to nocodazole, as well as with the subsequent washouts of nocodazole. The effect of FBXW5 knockdown on cell adhesion was also determined. As compared to the non-targeting control, which remained highly adherent, FBXW5 depleted MKN1 cells exhibited a ~60% decrease in cellular adherence (*p*-value = 0.006) (Figure 2I).

### 2.4. Knockout of FBXW5 Reduces Cellular Traction Forces

We next studied the role of FBXW5 in the generation of cellular traction forces. FBXW5 expression was stably knocked out in MKN1 cells by CRISPR/Cas9 transfection (Figure 3A). Here, cellular traction forces generated was assessed using a semi-quantitative application, which involves the use of fluorescent-labeled micro beads covalently bound on silica hydrogels. Quantification of cellular traction forces showed that knockout of FBXW5 in MKN1 cells resulted in a sustained decrease in the force exerted per area (*p*-value = 8.23 × 10^−210^) (Figure 3B), while overexpression of FBXW5 in MKN1 cells resulted in a sustained increase in the force exerted per area (*p*-value = 3.96 × 10^−182^) (Figure 3C). Subsequently, to examine the role of FBXW5 in turnover of adhesion force, the autocorrelation of the force magnitudes measured per area over some time was determined. Here, a steeper decrease in the resulting force autocorrelation functions was observed in MKN1 control cells as compared to MKN1 FBXW5 KO (knockout) cells (Figure 3D) and in MKN1 FBXW5 OE cells as compared to MKN1 control cells (Figure 3E). Lastly, based on quantification of autocorrelation function halftimes, we observed a significant increase in the halftimes with the knockout of FBXW5 (*p*-value = 0.0292) (Figure 3F), but a significant reduction in the halftimes with the overexpression of FBXW5 (*p*-value = 0.0037) (Figure 3G).

### 2.5. FBXW5 Knockdown Inhibits Activation of FAK Signaling

The effect of FBXW5 knockdown on FAK activation was explored. Of note, FBXW5 knockdown inhibited the activation of FAK, as represented by the decrease in p-FAK (Tyr397) protein expression. Consequentially, a reduction in p-Src (Tyr 416) and p-ERK1/2 (Thr 202/Tyr 204) protein expressions was also observed even though the expression of their unphosphorylated native forms remained unchanged. A downstream effect of this was observed in c-Myc protein expression level, as shown by a marked reduction (Figure 4A). Conversely, overexpression of FBXW5 in MKN1 cells could rescue these effects, as shown by an increase in p-ERK1/2 protein expression (Appendix AA).

Since c-Myc degradation is also regulated by F-box/WD repeat-containing protein 7 (FBXW7), a widely studied F-box member with known tumor suppressor functions, we determined the correlation between FBXW5 and FBXW7 expressions in gastric cancer, both in-vivo and in-vitro. Here, a negative correlation between FBXW5 and FBXW7 expressions was established from the analysis of a TCGA dataset (*r* = −0.32, *p*-value = 1.83 × 10^−7^) (Figure 4B). Consistently, an increase in FBXW7 expression was observed in the MKN1 FBXW5 KD cells (*p*-value = 0.003) (Figure 4C), as against the non-targeting counterpart. However, overexpression of FBXW5 and knockdown of FBXW7 in MKN1 cells (Appendix AC) did not show a corresponding decrease in FBXW7 (Appendix AB) and increase in FBXW5 expressions (Appendix AD), respectively. The inverse correlation observed between FBXW5 and FBXW7 in the TCGA samples could partly be replicated in-vitro. Hence, the data suggest that the clinical correlation observed between FBXW5 and FBXW7 may not be just stochastic, but it might have occurred indirectly through co-regulators that function either together or independently on FBXW5 and FBXW7. A potential role of these interactions in contributing to a particular cellular phenotype, therefore, needs to be further studied.

We then investigated if FBXW5 regulates the auto-phosphorylation of FAK through its proteasomal degradation function, since F-box members are known to play a role in ubiquitin-dependent proteosomal degradation of target proteins. Here, treatment of cells with proteasomal inhibitor MG132 did not have any effect on the decrease in p-FAK expression observed in MKN1 FBXW5 KD cells (Figure 4D), hence suggesting a ubiquitination and proteasomal degradation-independent role of FBXW5 in the regulation of FAK auto-phosphorylation.

### 2.6. Overexpression of FBXW5 Promotes In Vivo Gastric Cancer Tumorigenesis

The role of FBXW5 in cell proliferation as suggested by our in-vitro observations was further supported by an overexpression model of FBXW5. MKN1 FBXW5 OE cells demonstrated a significant increase in in-vitro cell proliferation (*p*-values = 0.004, 0.002, respectively) (Figure 5A) and in-vivo tumor growth. Here, NOD-SCID mice subcutaneously transplanted with MKN1 FBXW5 OE cells exhibited a significant increase in tumor burden and overall tumor growth (*p*-value = 0.001), as compared to those transplanted with MKN1 control cells (Figure 5B,C and Appendix AA). In addition, immuno-histochemical analyses also revealed an increase in Ki67 expression in MKN1 FBXW5 OE tumor samples as compared to the control tumor samples (Figure 5D).

### 2.7. Down Regulation in In Vivo Gastric Cancer Metastasis Observed from Splenic Injection of FBXW5 Knockout Cells

Similarly, the role of FBXW5 in cell migration as elucidated from in-vitro studies was further substantiated by a knockout model of FBXW5, which demonstrated a significant down regulation in tumor metastasis to the liver. MKN1 control (*p*-value = 0.0046) and MKN1 FBXW5 KO (*p*-value = 0.003) cells with stable luciferase expression were generated using a lentiviral V5-Luciferase expression vector (Figure 5E, Appendix AB). Here, NOD-SCID mice intrasplenically transplanted with MKN1 FBXW5 KO (+Luc) cells exhibited a significant decrease in tumor metastasis to the liver, as represented by the significant decrease in luminescence signal detected (*p*-value = 0.0393) (Figure 5F,G) and the lack of tumor nodules formation on harvested liver samples (Figure 5H), as compared to those transplanted with MKN1 control (+Luc) cells. In addition, H&E immunostaining revealed the development of tumor metastasis (black arrows) in MKN1 control (+Luc) liver tissues, while tumor metastasis was minimal or undetectable in the MKN1 FBXW5 KO (+Luc) liver tissues (Figure 5I). Immuno-histochemical analyses also revealed the presence of more Ki67 positive tumor cells in the MKN1 control (+Luc) liver tissues as compared to that in MKN1 FBXW5 KO (+Luc) liver tissues (Figure 5J).

### 2.8. High FBXW5 Expression Is Associated with Advanced, Metastatic Tumors and High RhoA Signaling

Analysis of a microarray dataset downloaded from the Gene Expression Ominbus (GEO) database (GEO Accession number: GSE13861) showed that FBXW5 expression was higher in both intestinal (*p*-value = 6.39 × 10^−4^) (Figure 6A, left panel) and diffuse types (*p*-value = 0.001) (Figure 6A, right panel) gastric cancer tissues as compared to healthy gastric tissues.

Analysis of a second independent microarray dataset (GEO Accession number: GSE103236) revealed an elevated expression of FBXW5 (*p*-value < 0.001) and ROCK1 (*p*-value = 0.046) in advanced stage gastric cancer tissues (M1) as compared to early stage gastric cancer samples (M0) and adjacent healthy gastric tissues (Figure 6B,C). Gene Set Enrichment Analysis (GSEA) was carried out between low FBXW5 samples and high FBXW5 samples, whereby the high FBXW5 samples were found to be enriched in gene sets related to metastasis (*p*-value < 0.001) (Figure 6D), RhoA signaling (*p*-value < 0.001) (Figure 6E) and advanced stage gastric cancer (*p*-value < 0.001) (Figure 6F, left panel). Concomitantly, the low FBXW5 samples were enriched in a gene set that was down regulated in advanced stage versus early stage gastric cancer (*p*-value < 0.001) (Figure 6F, right panel). Collectively, these analyses substantiated the role of FBXW5 in gastric cancer metastasis as elucidated from in-vitro and in-vivo studies.

Figure 7 is a graphical illustration of the proposed mode of FBXW5-induced regulation of intracellular signaling in gastric cancer.

## 3. Discussion

F-box proteins have garnered increasing attention for their role in cancer with emerging evidence suggesting dysregulation of F-box-mediated proteolysis as a cause of various human malignancies [17,26]. Albeit these, the role of FBXW5 in oncogenic signaling and tumorigenesis remains elusive. The current study provides a novel insight into FBXW5-induced tumorigenesis and metastasis mediated via the FAK-Src signaling pathway in gastric cancer.

The current study demonstrated that knockdown of FBXW5 significantly reduced in-vitro cell proliferation. Our study further reported changes in cell cycle profiles and an increase in cell apoptosis in the absence of FBXW5. Here, the increase in G0/G1 phase cells and a decrease in S phase cells could account for the decrease in in-vitro cell proliferation. Moreover, the opposing changes observed in caspase-3 activity, p21 and survivin protein expressions observed with the knockdown of FBXW5 added further credence to the anti-apoptotic role of FBXW5. Conversely, overexpression of FBXW5 significantly promoted in-vitro cell proliferation and in-vivo tumor growth. When considered together, our findings confirmed the tumorigenic potential of FBXW5 in gastric cancer and support a previous observation in NSCLC that suggested the role of FBXW5 in promoting cell proliferation and tumor growth [20].

Cell migration and adhesion are key processes underlying tumor metastasis, which involves the dynamic rearrangement of the actin cytoskeleton and regulation of focal adhesions turnover [22,23,27]. The present study demonstrated that loss of FBXW5 decreased in-vitro cell migration as supported by the decrease in the magnitude of traction forces. Our study provides primary evidence that there is significant depolymerization of actin stress fibers with cells possessing substantially less F-actin relative to G-actin in the absence of FBXW5. These findings validated that FBXW5 has a role in regulating F-actin assembly and/or maintenance. Alongside, unlike the normal occurrence of focal adhesion formation and disassembly in the non-targeting control cells, there is a reduction in the formation and turnover of focal adhesions in the absence of FBXW5, which contributes to reduced cell adhesion and cell migration [23], respectively. These findings are further substantiated by the slower rate of decay, which correlates directly with the rate of change in forces applied through the adhesions [28], as well as the increase in autocorrelation function halftimes.

Rho proteins mediate F-actin polymerization by binding to ROCK1, which phosphorylates MLC on the serine-19 residue [29,30,31] to coordinate the formation of stress fibers [32,33], and focal adhesions [13,34,35,36]. Thus, concomitant to the down regulated RhoA, ROCK1, and pMLC2 protein expressions in the absence of FBXW5, a decrease in actin stress fibers and focal adhesions was observed. Similarly, the activation of Cdc42 is also known to promote the assembly of focal adhesion complexes and the formation of filopodia to facilitate cell migration [34,37,38]. The reduction in Cdc42 activity, due to the knockdown of FBXW5, thus also accounts for the decrease in focal adhesion formation and cell migration. Consistently, knockout of FBXW5 significantly down regulated in-vivo tumor metastasis.

A study previously reported that induction of RhoA-ROCK1 signaling increases focal adhesion formation, which leads to the activation of FAK, which in turns regulates the organization of stress fibers, formation and turnover of focal adhesions [13,27,39]. Our findings fall in line with the previous observation, and further, suggest that down regulation of RhoA-ROCK1 and FAK activation in FBXW5 depleted cells could together contribute to the decrease in focal adhesions and migratory characteristics of gastric cancer cells. The aberrant activation of the FAK signaling pathway has been reported to confer the activation of ERK1/2 signal transduction to enhance cancer cell growth [40]. In the present study, FBXW5 knockdown cells with reduced p-FAK expression also displayed a down regulation in p-Src and p-ERK1/2 expression levels. The oncogenic potential of FAK activation is well substantiated by various studies that showed a role of FAK signaling in tumorigenesis, cell cycle progression at G1 phase [41], in cancer metastasis [11,42] and the inhibition of caspase-3 mediated apoptosis [43,44,45]. Collectively, our findings suggest that through the abovementioned cascade of events, FBXW5 has an oncogenic role in promoting gastric cancer tumorigenesis and metastasis.

The oncogenic role of FBXW5 is further supported by in-silico analysis of microarray dataset GSE13861 as higher FBXW5 expression was observed in both intestinal and diffuse-type gastric cancer tissues as compared to healthy gastric tissues. In addition, in-silico analysis of another microarray dataset GSE103236 revealed differential FBXW5 and ROCK1 expressions between early and advanced stage gastric cancer tissues, whereby expressions of both FBXW5 and ROCK1 are elevated in advanced stage gastric cancer samples as compared to early stage gastric cancer samples and adjacent healthy tissues. This set of analysis further validated our in-vitro findings that showed a reduction of ROCK1 expression in FBXW5 knockdown cells. Furthermore, GSEA analyses carried out between low and high FBXW5 samples supported the pro-metastatic role of FBXW5 as high FBXW5 samples were found to be enriched in gene sets related to metastasis, RhoA signaling and advanced stage gastric cancer. Collectively, these in-silico analyses added further credence to the oncogenic and pro-metastatic role of FBXW5 as elucidated from our in-vitro and in-vivo experimental findings.

## 4. Materials and Methods

### 4.1. Cell Culture and Drug Treatment

Gastric cancer cell lines CLS145, MKN1 (obtained from DUKE NUS, Singapore, Singapore), AGS and SNU1 (purchased from American Type Culture Collection) were maintained in RPMI-1640 (Nacalai Tesque, Kyoto, Japan) supplemented with 10% (*v*/*v*) heat inactivated HyClone^TM^ fetal bovine serum (FBS) (GE Healthcare, Chicago, IL, USA), 1% (*v*/*v*) penicillin/streptomycin (Thermo Fisher Scientific, Waltham, MA, USA) at 37 °C, 5% CO_2_ in an incubator. MKN1 cells were treated with 20 µM of MG132 as per previously published protocol [46].

### 4.2. Genomic Editing Approaches

To knockdown FBXW5, cells were transfected with either ON-TARGETplus Human FBXW5 siRNA–SMARTpool) or ON-TARGETplus Non-targeting Pool (Dharmacon, Lafayette, CO, USA) using DharmaFECT transfection reagent 1 (Dharmacon) according to the manufacturer’s protocol. For the generation of stable FBXW5 overexpression lines, cells were transfected with either Myc-DDK-tagged-Human F-box and WD repeat domain containing 5 (FBXW5) cDNA clone or Myc-DDK-tagged-pCMV6-Entry vector (OriGene, Rockville, MD, USA) using jetPRIME transfection reagent (Invitrogen, Carlsbad, CA, USA) according to the manufacturer’s protocol, followed by 100 ng/mL neomycin (Sigma-Aldrich, St. Louis, MO, USA) selection. For the generation of stable FBXW5 knockout lines, cells were transfected with either Human FBXW5 CRISPR/Cas9 KO plasmid or control CRISPR/Cas9 plasmid (Santa Cruz Biotechnology, Dallas, TX, USA) using UltraCruz^®^ transfection reagent (Santa Cruz Biotechnology) according to the manufacturer’s protocol, followed by sorting for top 10% of green fluorescent protein (GFP) positive cells by the fluorescence-activated cell sorting (FACS) method. For lentiviral particle production, HEK293T cells were co-transfected with Lentiviral V5-Luciferase expression vector (pLenti PGK Blast V5-LUC) (Addgene, Watertown, MA, USA) using jetPRIME transfection reagent (Invitrogen). Supernatants were then harvested 48 h after transfection. For the generation of stable luciferase-tagged lines, cells were infected with viral particles, followed by 0.25 μg/mL blasticidin (Thermo Fisher Scientific) selection.

### 4.3. Quantitative Real-Time PCR (qPCR) Analysis

Total RNA was extracted using the RNeasy Mini kit (Qiagen, Hilden, Germany), with the use of QIAshredder spin column for homogenization and an on-column DNase digestion. Using the Maxima First Strand cDNA Synthesis Kit (Thermo Fisher Scientific), 1 µg of the total RNA was reversely transcribed. The cDNA obtained was analyzed quantitatively using PrecisionFAST qPCR Master Mix (PrimerDesign, Southampton, UK) on an ABI 7500 Fast Real-time PCR system (Applied Biosystems, Foster City, CA, USA). The primers used are listed in Appendix A. Ct values were generated using default analysis settings and the housekeeping gene GAPDH was used as an internal control. Relative quantification (RQ) was calculated using the 2 ^−ΔΔCT^ method.

### 4.4. Protein Extraction and Western Blot Analysis

Cells were lysed in CelLytic buffer (Sigma-Aldrich), supplemented with Pierce protease inhibitor (Thermo Fisher Scientific) and PhosSTOP phosphatase inhibitor (Roche, Basel, Switzerland). Protein concentrations were measured by Bradford assay (Bio-Rad, Hercules, CA, USA). A total of 20 µg of protein was electrophoretically separated on 8% or 12% sodium dodecyl SDS-PAGE. The primary and secondary antibodies used are listed in Appendix A. The signals were visualized by ECL reagent (Amersham^TM^ ECL Select/Prime Western Blotting Detection System; GE Healthcare), followed by exposure to chemiluminescence film (Amersham Hyperfilm^TM^ ECL; GE Healthcare). The Western blot analyses were repeated twice for each protein tested.

### 4.5. Cell Proliferation (BrdU) Assay

The growth inhibitory effect of FBXW5 knockdown and the growth promoting effect of FBXW5 overexpression on MKN1 cells were measured using the BrdU proliferation assay (Roche) at 12, 24, and 48 h post siRNA transfection and at 24 and 48 h post cell seeding, respectively, according to the manufacturer’s protocol. The reaction was quantified by measurement of absorbance at 370 nm, with reference wavelength set at 492 nm, using a spectrophotometeric microplate reader (Tecan, Männedorf, Switzerland). The absorbance values are directly correlated to the amount of DNA synthesis.

### 4.6. Cell Cycle Assay

Changes in cell cycle distribution were determined by fluorescence-activated cell sorting (FACS). MKN1 cells were subjected to overnight serum starvation in serum free RPMI medium. MKN1 cells were released back into complete RPMI medium for 4 h before cell seeding. Cells were harvested at 36 h post siRNA transfection and fixed in ice-cold 70% ethanol for 30 min. Following fixation, cells were resuspended in 1 x PBS, treated with RNase (5 μg·mL^−1^; Ambion^®^; Thermo Fisher Scientific) and stained with propidium iodide (50 μg·mL^−1^) (Life Technologies, Carlsbad, CA, USA) for 30 min at 37 °C. For each sample, a total of 10,000 events were analyzed for DNA content by flow cytometry on a BD^TM^ LSR II (BD Biosciences, San Jose, CA, USA). The cells in sub-G1, G0/G1, S and G2/M phases were quantified using FlowJo software (version vX 0.7) (BD, Franklin Lakes, NJ, USA).

### 4.7. Cell Apoptosis Assay

Cell apoptosis was detected by FITC (fluorescein isothiocyanate) Annexin V Apoptosis Detection Kit I (BD Biosciences), according to manufacturer’s instructions. The cell samples were analyzed at 24 h post siRNA transfection by flow cytometry on a BD^TM^ LSR II (BD Biosciences) equipped with FlowJo software (version vX 0.7) (BD).

### 4.8. Caspase-Glo 3/7 Assay

The effect of FBXW5 knockdown on caspase 3/7 activity was measured under serum starved condition using the Caspase-Glo 3/7 assay kit (Promega, Madison, WI, USA), according to the manufacturer’s protocol. Capase 3 activity was quantified at 24 h post siRNA transfection by luminescence measurement using a GloMax^®^-Multi Detection System (Promega).

### 4.9. Cell Migration Assay

Changes in rates of cell migration was analyzed by a Radius^TM^ 24-well cell migration assay kit (Cell Biolabs, San Diego, CA, USA), according to the manufacturer’s instructions. Cell migration was evaluated by images captured by an inverted fluorescence microscope (Olympus, Shinjuku, Japan) at 0, 24, and 48 h post Radius^TM^ Gel removal.

### 4.10. Cell Adhesion Assay

Cell adhesion assay was performed from a protocol adapted from McClay et al., 1982 [47]. Cells were harvested at 36 h post siRNA transfection, and 1.5 × 10^5^ cells/well were reseeded onto 24-well plates. 3 h post cell reseeding; the 24-well plate was centrifuged twice in an inverted position at 2000 rpm for 2 min. Cells that were still adhered to the wells were fixed in 4% formaldehyde for 10 min. Following that, cells were stained with methylene blue for 10 min then rinsed thrice with 500 μL of 1× PBS with shaking for 5 min. Lastly, the number of cells that remained adhered to the plate was counted. Adhesion was defined as the percentage of plated cells that remained after centrifugation.

### 4.11. Immunofluorescence Staining Assay and Confocal Microscopy

MKN1 cells (1.5 × 10^5^ cells/well) were seeded onto coverslips in a 6-well plate. At 36 h post siRNA transfection, cells were fixed in 4% formaldehyde for 15 min and washed thrice in PBS. Fixed cells were permeabilized in 0.5% Triton X-100 (Sigma-Aldrich) for 5 min and blocked in blocking buffer (5% FBS (GE Healthcare), 0.3% Triton X-100 (Sigma-Aldrich) in PBS) for 1 h. Following that, cells were incubated with 1: 100 dilution of 546 Phalloidin (Thermo Fisher Scientific) for 2 h at room temperature in the dark. The nuclei were then labelled with DAPI (Thermo Fisher Scientific). Finally, fluorescence confocal images of cells were captured using a laser scanning confocal microscope (Leica TCS SP5) (Leica, Wetzlar, Germany).

On the other hand, MKN1 control and FBXW5 OE cells (2 × 10^5^ cells/well) were seeded onto coverslips in a 6-well plate. After overnight incubation, cells were processed according to the above protocol.

Similarly, MKN1 cells (1.5 × 10^5^ cells/well) were seeded onto coverslips in a 6-well plate. A nocodazole assay was performed at 24 h post siRNA transfection, in which, cells were exposed to one of the three conditions—(1) 0.025% DMSO in RPMI medium for 4 h, (2) 10 µM nocodazole in RPMI medium for 4 h or (3) 4 h 10 µM nocodazole followed by a 2 h washout with 0.025% DMSO in RPMI medium. After treatment, cells were fixed in 4% formaldehyde for 15 min, and washed thrice in PBS. Fixed cells were permeabilized in 0.5% Triton X-100 (Sigma-Aldrich) for 5 min and blocked in blocking buffer (5% FBS (GE Healthcare), 0.3% TritonX-100 (Sigma-Aldrich) in PBS) for 1 h. Following that, cells were incubated with 1: 100 dilution of anti-phospho-paxillin (Tyr118) (Cell Signaling, Danvers, MA, USA) overnight at 4 °C in the dark, followed by 2 h incubation with 1:100 dilution of Alexa Fluor 488 goat anti-Rabbit IgG (H+L) (Invitrogen) at room temperature. The nuclei were labeled with ProLong^TM^ Gold Antifade Mountant with DAPI (Thermo Fisher Scientific). Finally, fluorescence confocal images of cells were captured using a laser scanning confocal microscope (Leica TCS SP5) (Leica).

### 4.12. G-Actin/F-Actin In Vivo Assay

The amount of G-actin and F-actin in MKN1 NT and MKN1 FBXW5 KD cells was quantified using a G-actin/F-actin in-vivo assay kit (Cytoskeleton, Denver, CO, USA) according to the manufacturer’s instructions. Cells were harvested at 36 h post siRNA transfection and lysed in pre-warmed lysis/F-actin stabilizing buffer supplemented with protease inhibitor and ATP. The cell lysate was centrifuged for 5 min at 350× *g* to remove the cell debris. Cell lysate (100 µL) was ultra-centrifuged at 30,000 rpm for 1 h at 37 °C to pellet the F-actin with G-actin remaining in the supernatant. F-actin in the pellet was resuspended in 100 μL of F-actin depolymerization buffer on ice for 1 h with frequent pipetting. Equal volumes of G-actin and F-actin fractions were mixed with 5× SDS sample buffer and ran on SDS-PAGE. Western blot analysis was performed using the anti-actin primary and anti-rabbit HRP secondary antibodies provided in the kit.

### 4.13. Traction Force Microscopy

The traction force measurements were carried as previously described [48]. Briefly, CyA, and CyB components (Dow Corning) were mixed at 1:1 ratio and spin-coated at 800 rpm for 5 min on a Petri dish and cured at 80 °C for 2 h to achieve a flat substrate of height ~60–100 µm and stiffness 10–20 kPa. Beads were attached to the surface by salinizing it with 5% 3-aminopropyl trimethoxysilane (Sigma-Aldrich) in ethanol for 5 min followed by incubation with 1:3000 carboxylated fluorescent beads (100 nm) (Invitrogen) in deionized water. The beads were passivated with 1× Tris (Sigma-Aldrich, USA) for 10 min and pure fibronectin (100 µg/mL) was incubated on the substrate for 1 h. Between each step, the samples were rinsed 3 times with 1× PBS. The cells were seeded and allowed to adhere and spread prior to imaging, performed on an Olympus IX81 inverted microscope with temperature and CO_2_ controls. To get reference images after the live imaging was finished, the cells were removed by adding SDS. Bead displacements (with respect to its resting state) acquired during experiments were measured with PIVlab [49] and converted to cell traction forces with an ImageJ plugin [50].

### 4.14. In-Vivo Tumorigenesis Model of Gastric Cancer

Six weeks old NOD-SCID mice (*n* = 5) were used to establish the mouse xenograft model. 1 × 10^6^ MKN1 control or MKN1 FBXW5 overexpression (OE) cells suspended in 200 µL Matrigel (BD Biosciences) were injected subcutaneously into the left and right flanks of each mouse, respectively. Tumor growth was monitored over the course of 16 days, whereby tumor diameters were measured using Vernier calipers, and recorded three times a week. Tumor volume (TV) was calculated according to the formula: TV = (*W^2^* × *L*)/2 mm^3^, whereby ‘*W*’ is the width, and ‘*L*’ is the length of each tumor. The experiments were performed in accordance with the Institutional Animal Care and Use Committee (IACUC) guidelines. Mice were sacrificed if they had a tumor greater than 1.5 cm in diameter, if total tumor burden was greater than 10% of body weight if the tumor becomes ulcerated or interfered with mobility.

### 4.15. In-Vivo Metastasis Model of Gastric Cancer

Nine weeks old NOD-SCID mice (n = 5) were used to establish the in-vivo metastasis model. Firstly, the spleen was exteriorized via a left lateral flank incision. Subsequently, 1.5 × 10^6^ MKN1 control (+Luc) or MKN1 FBXW5 KO (+Luc) cells suspended in 15 µL of Matrigel (BD Biosciences) were injected directly into the spleen of each mouse, respectively. The peritoneum and skin were then closed in a single layer with surgical thread. Tumor metastasis from the spleen to the liver was monitored twice per week via bioluminescence imaging using the IVIS^®^ Spectrum in-vivo imaging system (PerkinElmer^®^) over the course of two weeks. Mice were sacrificed at two weeks post splenic injection. Bioluminescence signal was quantified using the Living Image^®^ software (PerkinElmer^®^). Liver samples were harvested from each sacrificed mouse for H&E staining. The experiments were performed in accordance with the Institutional Animal Care and Use Committee (IACUC) guidelines.

### 4.16. Analysis of Microarray Datasets from TCGA Database

Expression correlation analysis of FBXW5 and FBXW7 was performed using a TCGA gastric adenocarcinoma data set (https://tcga-data.nci.nih.gov/docs/publications/stad_2014/).

### 4.17. Analysis of Microarray Datasets from GEO Database

A microarray dataset for adenocarcinoma gastric cancer, downloaded from GEO with accession number (GSE13861) containing healthy (*n* = 19), intestinal-type (*n* = 20) and diffuse-type (*n* = 31) samples, was utilized to analyze the levels of FBXW5 mRNA expression across the three types of gastric tissues. A second independent microarray dataset of gastric cancer with accession number (GSE103236) containing pairwise adjacent healthy (*n* = 9) and tumor tissues (*n* = 8) was employed to evaluate the levels of FBXW5 and ROCK1 mRNA expression across the three distinct tissues: Adjacent healthy, early tumor (*n* = 6), or advanced tumor (*n* = 3). In addition, GESA analysis was performed between high-FBXW5 and low-FBXW5 samples to explore the correlations between FBXW5 expression and genes involved in RhoA signaling, metastasis and advanced tumors.

### 4.18. Statistical Analysis

Two-tailed Student’s *t*-test was used for differential comparison between two groups. A correlation test was performed to evaluate the strength of association between two continuous variables. *p*-value < 0.05 was considered statistically significant.

### 4.19. Ethics Statement

The study for animal experiments was approved by the National University of Singapore, Institutional Animal Care and Use Committee (NUS IACUC), protocol number: R17-0955(A)18.

## 5. Conclusions

In conclusion, the current study provides a detailed characterization of the oncogenic potential of FBXW5 in gastric cancer. In addition to confirming the tumorigenic role of FBXW5, our study identified a novel pro-metastatic role of FBXW5 in gastric cancer. Furthermore, we uncovered a previously unreported role of FBXW5 in regulating FAK-Src signaling. This study draws limitations in exploring the direct interacting partners of FBXW5. Nevertheless, our findings underscore a contributory role of FBXW5 to an aggressive tumor phenotype in gastric cancer and warrant further validation regarding its potential as a therapeutic strategy.

## Figures and Tables

**Figure 1 cancers-11-00836-f001:**
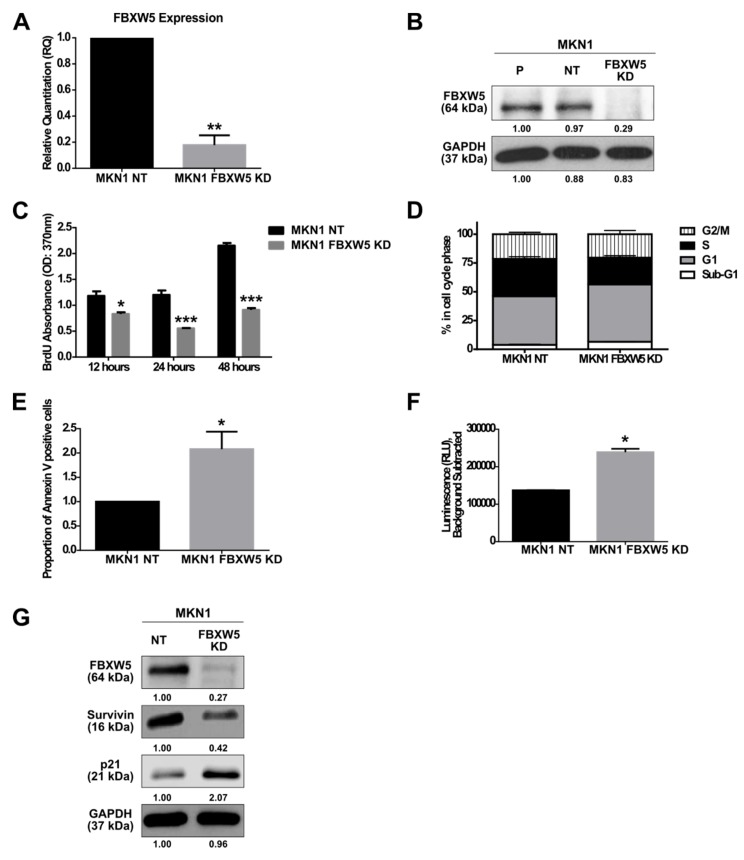
(**A**) Quantitative real-time PCR analysis of F-box/WD repeat-containing protein 5 (FBXW5) mRNA expression in MKN1 NT (non-targeting) and MKN1 FBXW5 KD (knockdown) cells (** *p*-value = 0.002). Values represent the average of three independent experiments and error bars denote standard deviations. At least three independent experiments were performed. (**B**) Western blot analysis of FBXW5 protein expression in MKN1 cells (P: Parental; NT: Non-targeting; FBXW5 KD: FBXW5 knockdown variant of MKN1). GAPDH served as the loading control. At least three independent experiments were performed. (**C**) Proliferation rates of MKN1 NT and MKN1 FBXW5 KD cells analyzed by (BrdU) proliferation assay across three time points (*p*-values = 0.01, 0.0008, 1.74 × 10^−6^, respectively). Values represent the average of three independent experiments and error bars denote standard deviations. (**D**) Cell cycle analysis of MKN1 NT and MKN1 FBXW5 KD cells in each cell cycle phase (*p*-values = 0.02, 0.04, 0.01, and N.S., respectively). Values represent the average of three independent experiments and error bars denote standard deviations. At least three independent experiments were performed. (**E**) The induction of apoptosis in serum starved MKN1 NT and MKN1 FBXW5 KD cells was evaluated by flow cytometry via an Annexin V-FITC apoptosis assay. The percentage of Annexin V positive cells was calculated. Data are presented as the mean values of the fluorescent intensities from two independent experiments and error bars denote standard deviations (* *p*-value = 0.03). (**F**) Caspase-3 activity of serum starved MKN1 NT and MKN1 FBXW5 KD cells (* *p*-value = 0.03). Values represent the average luminescence intensities of two independent experiments and error bars denote standard deviations. (**G**) Western blot analyses of survivin and p21 protein expressions in MKN1 NT and MKN1 FBXW5 KD cells. GAPDH served as the loading control. At least three independent experiments were performed. * *p*-value ≤ 0.05, ** *p*-value ≤ 0.01, *** *p*-value ≤ 0.001.

**Figure 2 cancers-11-00836-f002:**
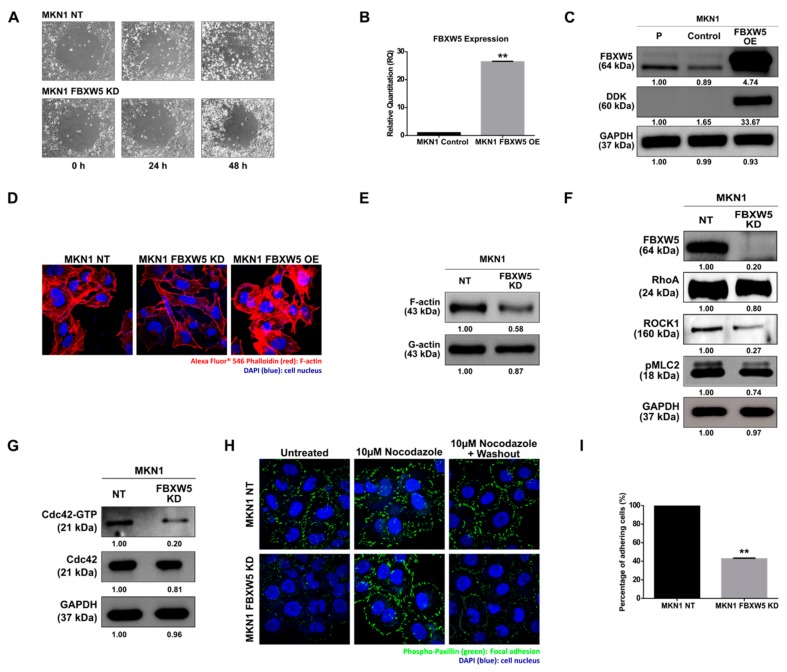
(**A**) Phase-contrast microscopic images (10×) of MKN1 NT and MKN1 FBXW5 KD cells, captured across three time points post removal of Radius Gel in the center of the well. At least two independent experiments were performed. (**B**) Quantitative real-time PCR analysis of FBXW5 mRNA expression in MKN1 control and MKN1 FBXW5 OE (overexpression) cells (** *p*-value = 0.009). Values represent the average of three independent experiments and error bars denote standard deviations. At least three independent experiments were performed. (**C**) Western blot analyses of FBXW5 and DDK protein expression in MKN1 cells (P: Parental; FBXW5 OE: FBXW5 overexpression variant of MKN1). GAPDH served as the loading control. At least three independent experiments were performed. (**D**) Immunofluorescent detection of F-actin (60×, oil immersion) in MKN1 NT, MKN1 FBXW5 KD, and MKN1 FBXW5 OE cells. Red: Phalloidin 546 labelled F-actin; blue: DAPI stained cell nucleus. At least two independent experiments were performed. (**E**) Western blot analyses of F-actin and G-actin protein expressions in MKN1 NT and MKN1 FBXW5 KD cells. At least two independent experiments were performed. (**F**) Western blot analyses of FBXW5, RhoA, ROCK1, and pMLC2 protein expressions in MKN1 NT and MKN1 FBXW5 KD cells. GAPDH served as the loading control. At least two independent experiments were performed. (**G**) Western blot analyses of Cdc42–GTP and Cdc42 protein expressions in MKN1 NT and MKN1 FBXW5 KD cells. GAPDH served as the loading control. At least two independent experiments were performed. (**H**) Immunofluorescent detection of focal adhesions (60×, oil immersion) in MKN1 NT and MKN1 FBXW5 KD cells, under three different conditions. Green: Alexa Fluor 488 labelled phospho-paxillin; blue: DAPI stained cell nucleus. At least two independent experiments were performed. (**I**) Percentage of adherent MKN1 NT and MKN1 FBXW5 KD cells quantified by microscopic examination (** *p*-value = 0.006). A total of four different fields were counted, and the average cell number was determined. Values represent the average of two independent experiments and error bars denote standard deviations.

**Figure 3 cancers-11-00836-f003:**
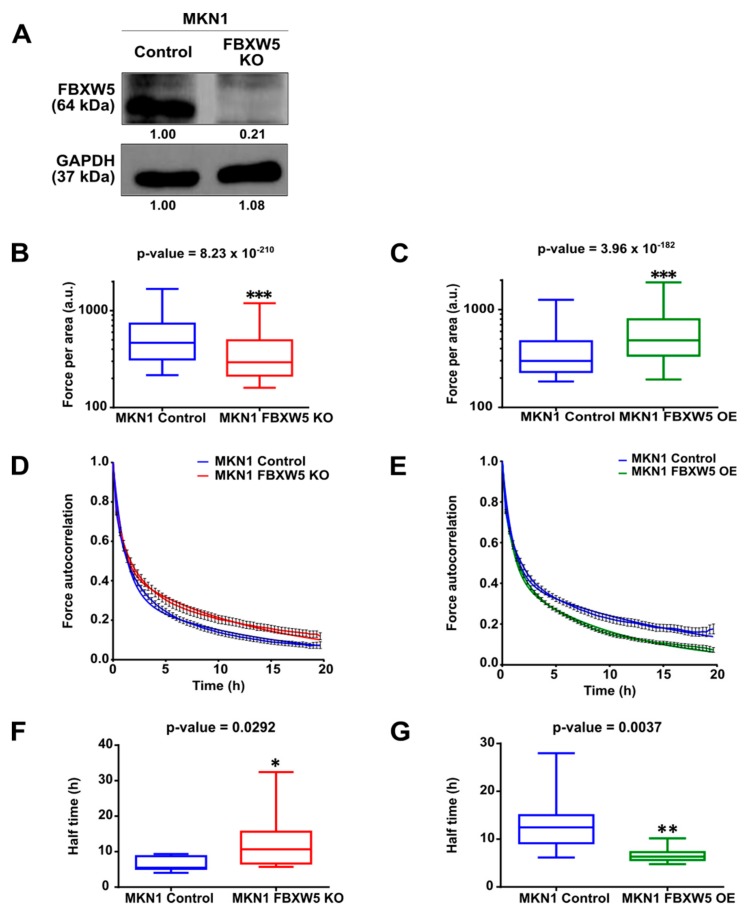
(**A**) Western blot analysis of FBXW5 protein expression in MKN1 control and MKN1 FBXW5 KO (knockout) cells. GAPDH served as the loading control. At least three independent experiments were performed. (**B**) Quantification of cellular traction forces of MKN1 control and MKN1 FBXW5 KO cells represented by box plots (*** *p*-value = 8.23 × 10^−210^). (**C**) Quantification of cellular traction forces of MKN1 control and MKN1 FBXW5 OE cells represented by box plots (*** *p*-value = 3.96 × 10^−182^). (**D**) Temporal dynamics of force application in MKN1 control and MKN1 FBXW5 KO cells plotted as mean ± SEM autocorrelation coefficient of the magnitude of forces per area over time. (**E**) Temporal dynamics of force application in MKN1 control and MKN1 FBXW5 OE cells plotted as mean ± SEM autocorrelation coefficient of the magnitude of forces per area over time. (**F**) Halftimes of fitted autocorrelation functions in MKN1 control and MKN1 FBXW5 KO cells represented by box plots (* *p*-value = 0.0292). (**G**) Halftimes of fitted autocorrelation functions in MKN1 control and MKN1 FBXW5 OE cells represented by box plots (** *p*-value = 0.0037).

**Figure 4 cancers-11-00836-f004:**
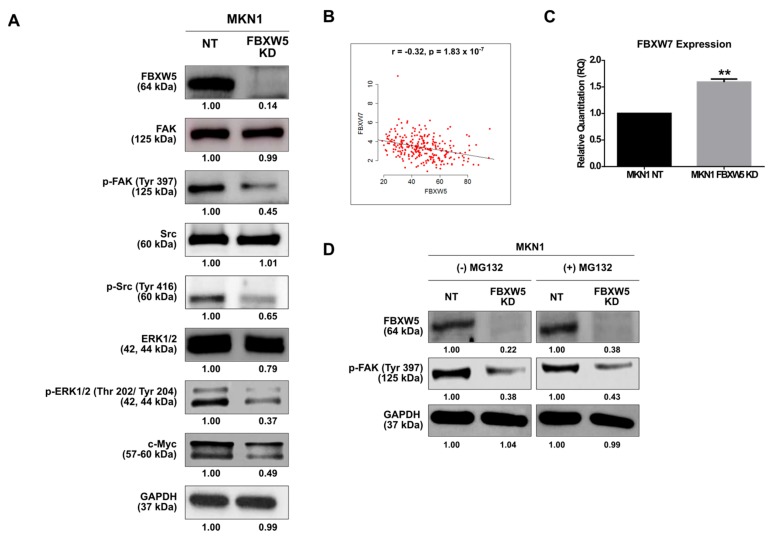
(**A**) Western blot analyses of expressions of proteins involved in the focal adhesion kinase (FAK) signaling pathway in MKN1 NT and MKN1 FBXW5 KD cells. GAPDH served as the loading control. At least three independent experiments were performed. (**B**) Correlation between FBXW5 and F-box/WD repeat-containing protein 7 (FBXW7) expressions. Matched FBXW5 and FBXW7 expressions of 262 primary gastric adenocarcinoma tissues in the TCGA data portal (*r* = −0.32, *p*-value = 1.83 × 10^−7^). (**C**) Quantitative real-time PCR analysis of FBXW7 mRNA expression in MKN1 NT and MKN1 FBXW5 KD cells (** *p*-value = 0.003). Values represent the average of three independent experiments and error bars denote standard deviations. (**D**) Western blot analysis of p-FAK protein expression in MKN1 NT and MKN1 FBXW5 KD cells, without and with the addition of MG132. GAPDH served as the loading control. At least three independent experiments were performed.

**Figure 5 cancers-11-00836-f005:**
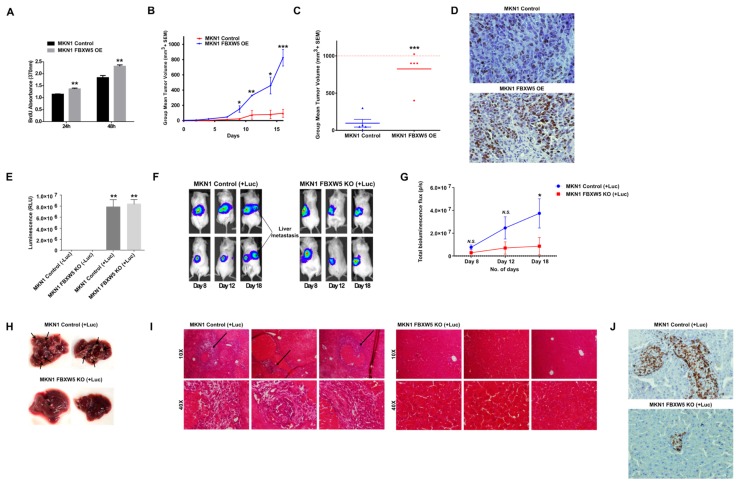
(**A**) Proliferation rates of MKN1 control and MKN1 FBXW5 OE cells analyzed by BrdU proliferation assay across two time points (*p*-values = 0.004, 0.002, respectively). Values represent the average of two independent experiments and error bars denote standard deviations. (**B**) Mean tumor volume (mm^3^) of MKN1 control and MKN1 FBXW5 OE cells at various indicated days over the course of 16 days (*p*-values = 0.035, 0.009, 0.022, 0.001, respectively). (**C**) Mean tumor volume (mm^3^) of MKN1 control and MKN1 FBXW5 OE cells at the end of 16 days (*** *p*-value = 0.001). (**D**) Immunohistochemical analysis of Ki67 expression in MKN1 control (top) and MKN1 FBXW5 OE (bottom) xenograft tissues. (**E**) Level of luminescence (RLU) of MKN1 control (−/+ Luc) (** *p*-value = 0.0046) and MKN1 FBXW5 KO (−/+ Luc) (** *p*-value = 0.003) cells. (**F**) Representative images of tumor metastasis in NOD-SCID mice intrasplenically injected with MKN1 control (+Luc) or MKN1 FBXW5 KO (+Luc) cells. Five animals were randomly assigned to each group and observed over the course of 18 days, with bioimaging performed on day 8, 12 and 18. (**G**) Quantification of bioluminescence imaging signal intensity in NOD-SCID mice intrasplenically injected with MKN1 control (+Luc) or MKN1 FBXW5 KO (+Luc) cells over the course of 18 days (* *p*-value = 0.0393). (**H**) Representative images of liver metastasis harvested from NOD-SCID mice intrasplenically injected with MKN1 control (+Luc) or MKN1 FBXW5 KO (+Luc) cells. Five animals were randomly assigned to each group and mice were sacrificed on day 18. (**I**) Representative images of haematoxylin and eosin (H&E) immunostaining of liver tissue sections prepared from NOD-SCID mice intrasplenically injected with MKN1 control (+Luc) or MKN1 FBXW5 KO (+Luc) cells. (**J**) Immunohistochemical analysis of Ki67 expression of liver tissue sections prepared from NOD-SCID mice intrasplenically injected with MKN1 control (+Luc) (top) or MKN1 FBXW5 KO (+Luc) (bottom) xenograft tissues. * *p*-value ≤ 0.05, ** *p*-value ≤ 0.01; *** *p*-value ≤ 0.001.

**Figure 6 cancers-11-00836-f006:**
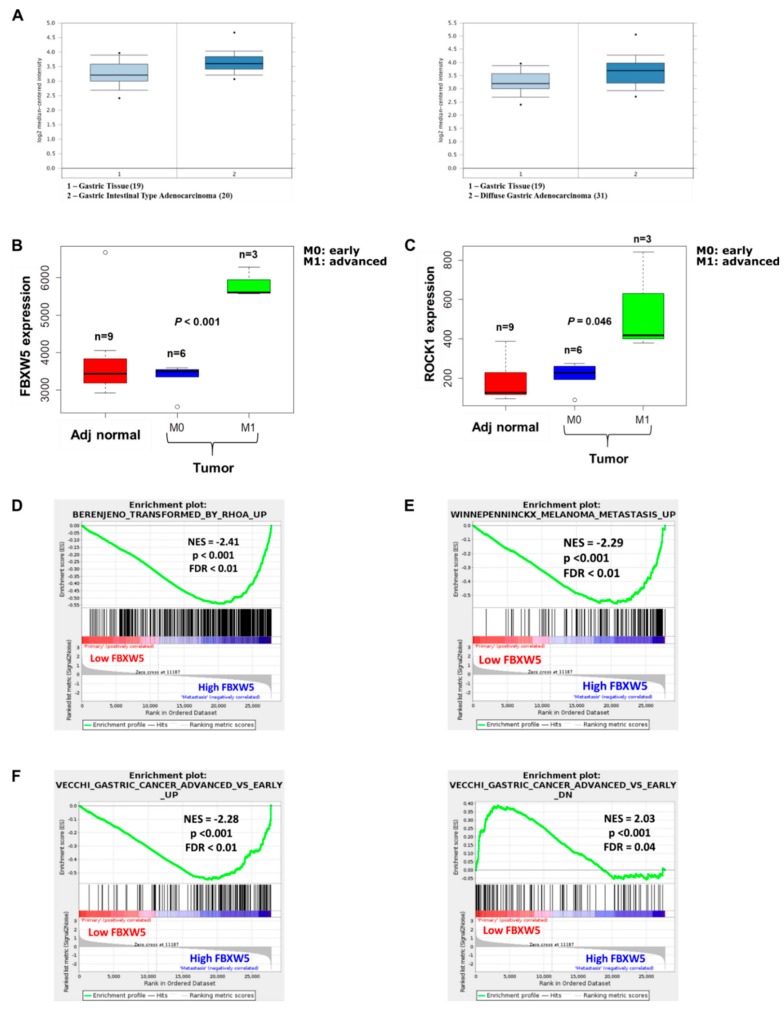
(**A**) FBXW5 expression level across healthy gastric tissues, intestinal-type (left panel) and diffuse-type (right panel) tumor tissue samples (gastric cancer: GSE13861). (**B**) FBXW5 expression across adjacent healthy, early, and advanced tumor tissues (gastric cancer: GSE103236). (**C**) ROCK1 expression across adjacent healthy, early, and advanced tumor tissues (gastric cancer: GSE103236). (**D**) GSEA plot showing enriched RhoA signaling in high FBXW5 tumors compared to low FBXW5 tumors. (**E**) GSEA plot showing enriched metastasis in high FBXW5 tumors compared to low FBXW5 tumors. (**F**) GSEA plot showing enriched advanced tumors in high FBXW5 tumors (left panel) and enriched early tumor in low FBXW5 tumors (right panel).

**Figure 7 cancers-11-00836-f007:**
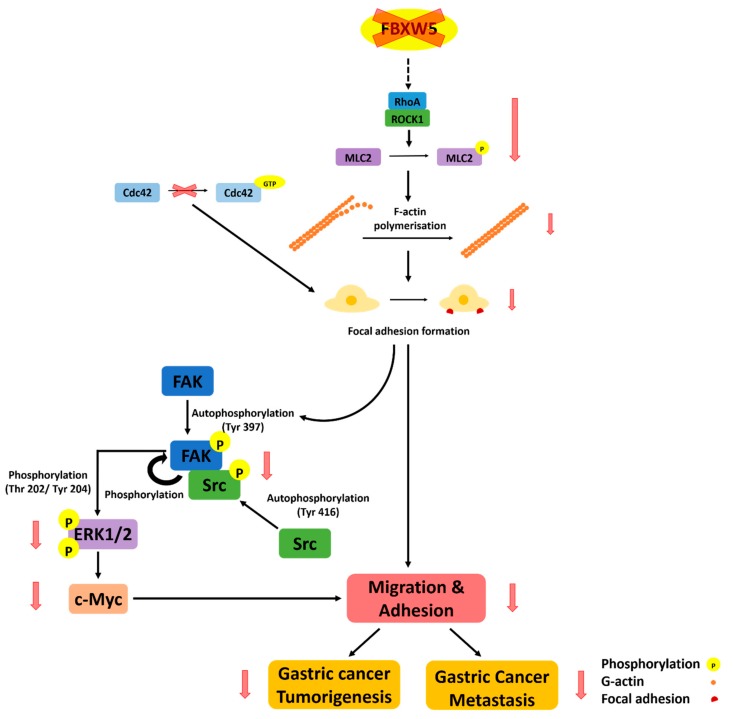
A graphical illustration of the proposed mode of FBXW5-induced regulation of intracellular signaling in gastric cancer.

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
