# Peer review of "FBXW5 Promotes Tumorigenesis and Metastasis in Gastric Cancer via Activation of the FAK-Src Signaling Pathway"

_cancers, 2019, doi:10.3390/cancers11060836_

Round 1
Reviewer 1 Report
This manuscript reports the role of FBXW5, a ubiquitin E3 ligase, in the pathogenesis of gastric cancer. In general, the experiments are well designed and data support the conclusion the authors have drawn. My major concern is that the authors used only one cell line for data generation. It is reasonable to use the cell line that has a higher level of FBXW5 for the knockout experiments. However, for FBXW5 overexpression, it would be better to use low FBXW5 expression cell lines such as AGS. Data including two different cell lines would be more convincing. In addition, liver metastasis analysis that was based on intrasplenic injection of cancer cells may not represent an ideal model system, given gastric cancer research. The study also lacks the mechanistic investigation of FBXW5 in regulating FAK activity.
Author Response
Responses to comments from Reviewer 1:
Comment 1: For FBXW5 overexpression, it would be better to use low FBXW5 expression cell lines such as AGS. Data including two different cell lines would be more convincing.
Response 1: We agree with the reviewer’s comments. FBXW5 was also stably overexpressed in AGS cells. However, in-vivo xenografts of both AGS control and OE cells showed very slow growth dynamics in our mouse models.
Comment 2: Liver metastasis analysis that was based on intrasplenic injection of cancer cells may not represent an ideal model system, given gastric cancer research.
Response 2: We agree with the reviewer’s comments. A tail vein injection of cancer cells was also performed to assess metastasis. However, the cell lines failed to micro-metastasize in this model system.
Comment 3: The study also lacks the mechanistic investigation of FBXW5 in regulating FAK activity.
Response 3: We agree with the reviewer’s comments. Our future studies will focus on the molecular mechanisms by which FBXW5 regulates FAK signaling in gastric cancer cells. To delineate this, we intend to use NGS-transcriptome analysis of FBXW5 knock down cells and mass spectrometric analysis.

Reviewer 2 Report
The manuscript entitled, “FBXW5 promotes tumorigenesis and metastasis in gastric cancer via activation of the FAK-Src signaling pathway”, described a series of in vitro and in vivo studies that were conducted to determine the role of FBXW5 in the tumorigenesis and metastasis of gastric cancer using in vitro and in vivo models. The study design and results were well described. The authors used the in vitro models to demonstrate that knockdown of FBXW5 in MKN1 human gastric cancer cells significantly decreased cell proliferation, migration and adhesion, and cell cycle progression. The authors then employed an in vivo subcutaneous tumor xenograft model and metastatic tumor model to verify the oncogenic role of FBXW5. The authors conclude that the contribution of FBXW5 to an aggressive tumor phenotype in gastric cancer warrants further investigations regarding its potential as a therapeutic target.
Major points:
1. It was unclear why MKN1 FBXW5 KD and MKN1 FBXW5 OE cell lines were often used in different assays. Was it because the results were negative?
(1) Fig 2a and Supplementary Fig. 2a. presented the results of two different cell migration assays. The former involved MKN1 NT and MKN1 FBXW5 KD cells, and the latter involved MKN1 control and MKN1 FBXW5 OE cells. What was the reason that the same migration assay method was not used to test all four cell lines at the same time?
(2). MKN1 FBXW5 OE cells were used in the in vivo metastasis model but not in the subcutaneous tumor xenograft model, while MKN1 FBXW5 KD cells was used in the subcutaneous tumor xenograft model but not in the metastasis model. What was the rationale of not including both FBXW5 KD and FBXW5 OE cell lines in the same in vivo tumor model?
2. Besides Supplementary Fig. 4a, was the Western blotting assay done in tumor tissue samples to verify the in vitro observation of changes in certain important effector proteins described in Figure 7?
3. MKN1 control and MKN1 FBXW5 KD tumor samples collected from the in vivo metastasis model of gastric cancer should be subjected to the immunohistochemical analysis of Ki67 expression.
4. Is knockdown of FBXW5 associated with the EMT process?
Minor points:
1. Line 296-299: How many animals in the control MKN1 group developed liver metastasis? Was tumor growth at the splenic injection site analyzed in terms of tumor size and Ki67 expression?
2. Line 414: Was 5-FU used in any experiment?
Author Response
Responses to comments from Reviewer 2:
Comment 1: It was unclear why MKN1 FBXW5 KD and MKN1 FBXW5 OE cell lines were often used in different assays. Was it because the results were negative?
Response 1: MKN1 FBXW5 KD and MKN1 FBXW5 OE cell lines were used in different assays because for assays that required longer incubation times, cell lines with stable FBXW5 expression (MKN1 FBXW5 OE/ KO) were used instead of transient expressing ones (MKN1 FBXW5 KD). An MKN1 FBXW5 KO model was only used for traction force microscopy and in-vivo metastasis assay because MKN1 FBXW5 KO cell line was generated at a later point during the experiments.
Comment 1 (a): Fig 2a and Supplementary Fig 2a. presented the results of two different cell migration assays. The former involved MKN1 NT and MKN1 FBXW5 KD cells and the latter involved MKN1 Control and MKN1 FBXW5 OE cells. What was the reason that the same migration assay method was not used to test all four cell lines at the same time?
Response 1 (a): Cell migration rates between MKN1 Control and MKN1 FBXW5 OE cells was compared using the conventional wound healing assay. However, the same assay was not used for MKN1 NT and MKN1 FBXW5 KD cells. This is because different transfection protocols and controls were used to overexpress or knockdown FBXW5 in MKN1 cells. More cell death was observed after siRNA transfection to knockdown FBXW5. Thus, it was difficult to identify an area with confluent cells for the creation of the wound. We also tried to re-seed transfected cells onto a 6-well plate for a wound healing assay but it was also difficult to identify an area with confluent cells. So we decided to look for an alternative assay and the RadiusTM 24-well cell migration assay kit appeared as the best option with confluent cells around the ‘circular wound’.
Comment 1 (b): MKN1 FBXW5 OE cells were used in the in vivo metastasis model but not in the subcutaneous tumor xenograft model, while MKN1 FBXW5 KD cells was used in the subcutaneous tumor xenograft model but not in the metastasis model. What was the rationale of not including both FBXW5 KD and FBXW5 OE cell lines in the same in vivo tumor model?
Response 1 (b): FBXW5 KD cell line was not included in the in-vivo subcutaneous model because this experiment involved the long term monitoring of tumor growth. The knockdown of FBXW5 by siRNA was transient, whereas the overexpression of FBXW5 is stable in MKN1 cell lines. Furthermore, since in-vitro proliferation assay performed using FBXW5 KD cell line demonstrated a decrease in cell proliferation, we were interested to find out whether overexpression of FBXW5 will generate a corresponding increase in tumor growth. On the other hand, FBXW5 KO cell line was only generated at the later stage of the project, after the completion of the in-vivo subcutaneous assay.
FBXW5 OE cell line was not included in the in-vivo metastasis model because intrasplenic injection of both MKN1 Control and MKN1 FBXW5 OE cells gave rise to metastasis to the liver and there was no significant difference in the extent of liver metastasis between the two. This data was not included in the manuscript because the assay was performed in NSG mice using non-luciferase-tagged cell lines as a preliminary experiment.
Comment 2: Besides Supplementary Fig 4a, was the Western blotting assay done in tumor tissue samples to verify the in-vitro observation of changes in certain important effector proteins described in Fig 7?
Response 2: Western blotting data using tumor tissue samples to verify the in-vitro observation of changes in the important effector proteins described in Fig 7 were not obtained. Western blot analyses should have been performed. However, tumor tissues were exhausted after sourcing for IHC and optimization for Western blot analyses.
Comment 3: MKN1 Control and MKN1 FBXW5 KD tumor samples collected from the in-vivo metastasis model of gastric cancer should be subjected to the immunohistochemical analysis of Ki67 expression.
Response 3: Thank you for the suggestion. We have performed a Ki67 staining on the slides (liver tissue samples harvested from MKN1 Control (+Luc) and MKN1 FBXW5 KO (+Luc) mice) which were prepared for H&E staining previously. Immuno-histochemical analyses revealed the presence of more Ki67 positive tumor cells in the MKN1 Control (+Luc) liver tissues as compared to that in MKN1 FBXW5 KO (+Luc) liver tissues. The additional data has been included in the manuscript under Figure 5j. The manuscript has also been updated accordingly to include this finding (Lines 321 – 323 & Lines 306 – 307). (Please refer to attachment for the Ki67 staining images.)
Comment 4: Is knockdown of FBXW5 associated with the EMT process?
Response 4: We have performed a Western blot analysis to compare the protein expressions of vimentin, snail, E-cadherin and N-cadherin between MKN1 NT and MKN1 FBXW5 KD. However, we did not observe any significant difference in expression levels. (Please refer to attachment for the Western blot images.)
Comment 5: (Line 296-299) How many animals in the control MKN1 group developed liver metastasis? Was tumor growth at the splenic injection site analyzed in terms of tumor size and Ki67 expression?
Response 5: All five mice in the control MKN1 group developed liver metastasis. Analyses of tumor growth in terms of tumor size and Ki67 expression at the splenic injection site were not obtained. This is because tumor growth on the spleen was minimal even though cancer cells were injected into the spleen.
Comment 6: (Line 414) Was 5-FU used in any experiment?
Response 6: No. 5FU treatment was not performed in the scope of this manuscript. We have removed the sentence “5-FU (Pharmachemie BV) was stored at room temperature.” from the M&M section (Lines 437 – 438).

Round 2
Reviewer 1 Report
The authors addressed my questions by explanations without providing new data.
Reviewer 2 Report
The authors have addressed all of my concerns.